# A Methodology of Workshops to Explore Mutual Assistance Activities for Earthquake Disaster Mitigation

**DOI:** 10.3390/ijerph18073814

**Published:** 2021-04-06

**Authors:** Kazuki Karashima, Akira Ohgai

**Affiliations:** 1Department of Architecture, Maebashi Institute of Technology, Maebashi 3710816, Japan; 2Toyohashi University of Technology, Toyohashi 4418580, Japan; ohgai.akira.vd@tut.jp

**Keywords:** planning support tool, mutual assistance, earthquake, community-based activities, GIS, multi-agent system, community disaster management plan

## Abstract

To minimize the damage caused by large earthquakes, mutual assistance activities between residents and rescue victims (i.e., to support residents who cannot evacuate individually) are important. To enhance these activities, the technologies and methods for creating a Community Disaster Management Plan (CDMP), based on the quantitative evaluation of mutual assistance abilities, are required. However, the lack of a method for it is a key issue. This study aims to develop a methodology of workshops for making CDMPs by using the developed support tool by the authors to explore and promote mutual assistance activities. Through the demonstration and examination of a Community Disaster Management Plan on actual districts, the findings mentioned in this article were obtained. Moreover, the usability of this method is shown. In particular, this method is effective at revising CDMPs, and raising resident awareness on the importance of mutual assistance. The suggested method can also improve the lack of techniques involved in promoting mutual assistance.

## 1. Introduction

### 1.1. Background and Objective

There is an urgent need for communities to find ways to improve safety in built-up areas in order to minimize the damage caused by large earthquakes. Although improving built environments (e.g., reconstructing old buildings or widening narrow roads) is important, it is difficult, partially due to the huge costs of improvements and the complexity of consensus-building among stakeholders.

Therefore, improving built environments takes a long time.

Considering this issue, improving disaster response capability, enhancing self-help, mutual assistance, and public help for disaster mitigation, is significant for the safety of these built-up areas.

However, an issue of concern is the increasing ratio of the elderly population, who cannot evacuate without assistance from others. This issue is particularly serious in a shrinking nation, such as Japan. Another concern is that earthquake damage, such as fires and building collapses, occur simultaneously and in large numbers, just after a huge earthquake. Due to these issues, as well as the limitations of public help, it can be difficult to ensure efficient emergency response. Therefore, mutual assistance among local residents is extremely important.

In Japan, after the Great East Japan earthquake in 2011, it was re-recognized that a coordinated response (e.g., self-help, mutual assistance, and public help) is crucial after large-scale wide-area disasters [1]. In preparation for the occurrence of large-scale earthquakes (e.g., the Nankai megathrust earthquakes or the Tokyo Metropolitan area inland earthquake which will be forecasted in near future), disaster prevention measures that consider mutual assistance activities are imperative.

Based on these social situations, in Japan’s Disaster Countermeasures Basic Act, revised in 2013, regulations concerning self-help and mutual assistance were added. According to the regulation, the promotion of community-based activities for disaster mitigation (involving mutual assistance among residents) are required [2]. Consequently, the cabinet office amended the Disaster Countermeasures Basic Act in June 2013, and created the “Community Disaster Management Plan” (CDMP), a disaster management activities plan for businesses and residents of local communities [3].

According to the cabinet office, creating CDMPs is urged all over Japan [4]. However, there is no method to create a CDMP based on the quantitative evaluation of mutual assistance capability at a district level, to identify issues, or to encourage discussions to improve those issues.

This study aims to develop a methodology of workshops (WSs) for making CDMPs, and for raising resident awareness on using support tools that explore and promote mutual assistance activities at the district levels, based on the results of our previous study [5,6], utilizing the geographic information system (GIS) and multi-agent system (MAS). In addition, the usabilities and issues of the method for exploring CDMP are clarified through demonstration and examination of CDMP in an actual district.

In the Materials and Methods section, the outline of the developed tool (from our previous study) is explained. We also include revised points and a methodology of workshops (i.e., for making CDMPs and for raising resident awareness).

Then, in order to evaluate the functionality and the usability of the revised tool, a demonstration was observed in an actual district with vulnerability to earthquakes. Using the results, two hearing-based surveys were conducted to obtain information by local governments. One was the local government that was familiar with the district that was targeted for experimental use. The other was the local government that had experienced mutual assistance activities after the Great Hanshin-Awaji earthquake. The evaluation of the revised tool, the demonstration WSs (i.e., for evaluating the methodology of WSs for making CDMPs), to raise resident awareness, according to the proposed method, were examined.

After the WSs, questionnaire survey results were analyzed to evaluate the proposed method. Discussions took place throughout the analysis.

### 1.2. Previous Studies

There are many previous studies on earthquakes. To reduce the damage of an earthquake, damage simulations or disaster damage predictions are important. For example, Yoshii [7] and Lazaridou-Valotsos [8] showed the social impacts of earthquake predictions (an earthquake occurred in Greece in 1988). In addition, the expected effects of mutual assistance activities is important for this study. Kawata [9] showed—through the survey of the great Hanshin-Awaji earthquake—that over 75% of the people who were buried by the rubble were rescued by mutual assistance activities. Furthermore, the mutual assistance capability evaluation method is important. Akiyama et al. [10] and Ogawa et al. [11] suggested a method to quantitatively estimate the initial response capabilities via mutual assistance activities during huge earthquake disasters. The result of the method is provided at a large scale to analyze the vulnerability of mutual assistance activities from a city scale. In other words, there was no concrete method to evaluate mutual assistance capability at a district level or neighborhood community level. To improve this issue, Karashima and Ohgai [5,6] developed a method to evaluate mutual assistance capability at a district level or neighborhood community level.

Model development is another point related to the studies; it involves simulating human behavior, such as mutual assistance activities or evacuation responses to natural disasters. One popular method is utilizing the multi-agent system (MAS). For example, D’Orazio et al. [12] proposed an innovative approach for promoting the effect of evacuation to an earthquake, presenting an agent-based model to describe phases and rules of motion for pedestrians. Wagner and Agrawal [13] presented a prototype of a computer simulation and decision support system that uses agent-based modeling to simulate crowd evacuation in the presence of a fire disaster, and provides testing of multiple disaster scenarios at virtually no cost. Takabatake et al. [14] developed an agent-based tsunami evacuation model that considers the different behaviors of local residents and visitors, which can estimate the evacuation time, number of individuals reaching each evacuation area, the location of bottlenecks, and the number of casualties. Wang et al. [15] presented a multimodal evacuation simulation for a near-field tsunami through an agent-based modeling framework to investigate how the varying decision time impacts the mortality rate, and how the choice of different modes of transportation (i.e., walking and automobile) and existence of vertical evacuation gates impact the estimation of casualties. Mostafizi et al. [16] presented an agent-based modeling framework to evaluate vertical evacuation behavior and shelter locations for a near-field tsunami hazard from a Magnitude 9.0 Cascadia Subduction Zone earthquake. However, these studies have not considered mutual assistance activities.

Osaragi and Oki [17] developed a comprehensive simulation model that integrates property damages and various activities (rescue activity, firefighting activity, and wide-area evacuation activity) by local residents in the event of a large earthquake. In addition, Oki and Osaragi [18,19] developed a wide-area evacuation in densely built-up wooden residential areas, incorporating rescue activities by local residents, demonstrating the effectiveness of rescue activities by local residents. However, these are not for community-based activities to promote mutual assistance capability among residents.

The Japan Society of Community Disaster Management Plan (SCDMP) was established in June 2014 [3] (the SCDMP published the journal, C+Bousai). There are many studies on CDMP. For example, Inaba [20] showed the importance of social capital surveys for community disaster management plans. However, there is no study on the quantitative evaluation of mutual assistance that explores a CDMP.

Karashima and Ohgai [5,6] showed the necessity of a technique (as well as the fundamental development of a tool) that explores mutual assistance activities in community-based activities. However, the technique has not been able to conduct objective evaluations of the usabilities, or respond to the following issues:(a)The features that are difficult to cross, such as rivers, railroad tracks, and wide trunk roads were not considered for evaluating mutual assistance map.(b)The building and the disaster situation (fire spread and rubble generation) were not constructed using a grid cell model, and the evacuation route was not constructed using a network model. For this, precise evaluation is difficult.(c)The results of the fire spread/road blockage model are changed by each simulation. This made it impossible to perform a fixed simulation of the disaster situation.

This study is unique because of the method used for exploring a CDMP (by using a technique to quantitatively evaluate mutual assistance capability).

For the method, this study attempts to revise the tool by Karashima and Ohgai [5,6] to improve the issues mentioned above.

## 2. Materials and Methods

### 2.1. Examination of the Required Tool

A literature survey was conducted to organize the tool configuration and necessary functions. A representative reference book on improving densely built-up areas in Japan [21] shows a general method for exploring earthquake disaster countermeasures: (a)First, local governments or experts evaluate the vulnerability to earthquakes at the city level, and identify the areas that need improvement.(b)Next, the local government approaches the neighborhood association to improve the issues.(c)After consensus building, concrete examination to improve the vulnerability of the area is explored by the residents and local government.

In addition, to comprehend the required techniques to promote mutual assistance capability, Karashima and Ohgai [5] show that technology possessing the following functions is important in community-based activities for disaster mitigation:i.Quantitatively evaluate and analyze mutual assistance capability by visual representation.ii.Promote an understanding of the importance of mutual assistance among residents.iii.Support the examination of community-based activities, considering mutual assistance.

### 2.2. Structure of the Tool

Based on the results, a proposed tool consists of two sub-tools: (1) a GIS based tool for evaluating mutual assistance capability to extract the area showing low capability (GIS sub-tool), and (2) an MAS-based evacuation simulator that explores community-based activities in relation to mutual assistance, and illustrates the usability of the mutual assistance activities (MAS sub-tool) (see Figure 1).

Initially, experts, such as individuals in local governments, consultant staff, and researchers, evaluate mutual assistance capability in a wide area, such as an elementary school district unit. Using the GIS sub-tool, local governments can extract the area showing low capability. Then, local governments should promote the capability of the extracted area and facilitate community-based activities to promote capability. Therefore, using an MAS sub-tool, users, such as residents and experts, can explore community-based activities, considering mutual assistance, and visualize the usability of mutual assistance activities.

### 2.3. GIS Sub-Tool

#### 2.3.1. Evaluation Method

The evaluation method used for calculating mutual assistance capability was the method suggested by Akiyama et al. [10] and Ogawa et al. [11]. First, the expected value for the rescue of each person was calculated, refer to Table 1. The expected value for rescue is the numerical value showing the capability to rescue the victims (e.g., pulling a survivor from the wreckage) in accordance with gender, age, and strength. This table was organized by the Tokyo Fire Department [22] based on the actual rescue activities conducted during the Great Hanshin-Awaji earthquake in 1995 in Japan. For example, the expected value of a 40-year-old man is calculated by the following formula:strength (0.93) × executing rate (0.298) × activity rate (0.72) = 0.1995

The strength value is calculated in accordance with age and gender on the basis of the strength value of a man in his teens through his twenties; set at one. The value of executing rate is set in accordance with the condition of actual rescue activities conducted during the Great Hanshin-Awaji earthquake. The activity rate value is the ratio of residents who can perform rescue activities, considering the degree of daily activities.

The expected value *Rrj* of building *j* is calculated as the total residents’ expected value at building *j*. However, in this paper, elementary school students and junior high school students have no capability for rescue. Second, the expected value is weighted by distance based on the assumption that residents take some time to discover or recognize those persons who cannot evacuate without some assistance, in accordance with the distance. Therefore, the range limit in which residents can discover a person who cannot evacuate the building *i* is set at 100 m. The residents’ expected value is decreased with the increase of distance from building *i*. The weighted value *dwi* of building *j*, having *dj* (m) distance from building *i*, is calculated by Formula (1). The evaluation unit of the mutual assistance capability is the building unit, based on the assumption that it is easy for residents to understand the capability. Finally, mutual assistance capability is calculated by formula (2).
(1)$$\mathrm{Dw}j=\frac{\log\left(1+X\right)}{2\left(\log\left(1+Dj\right)+1\right)}\;\left(0\;\geq \;j\;\geq \;X\right)$$
(2)$$\mathrm{The}\;\mathrm{capability}\;\mathrm{of}\;\mathrm{mutual}\;\mathrm{assistance}\;=\;\frac{\sum Rrj\;\times dwj}{5}$$

#### 2.3.2. The Features That Are Difficult to Cross

Using the previous fundamental development, it was difficult to consider the features that are difficult to cross when evacuating or conducting mutual assistance activities. In this sub-tool, by considering features that are difficult to cross, such as rivers, railroad tracks, and wide trunk roads, evaluations that are closer to the actual situation can be performed (see Figure 2).

The method to improve the problem is by creating a function to designate the obstruction, such as rivers, railroad tracks, and wide trunk roads, when evacuating or conducting mutual assistance activities on the interface of the tool.

#### 2.3.3. Using an Interface Method

At first, users set the calculating range by a mouse operation after pushing button 1. By using the delete button, the user can set it again. After the operation, users can set some structures that residents cannot cross (e.g., railway tracks and rivers) by a mouse operation after pushing button 2—the same as mentioned above. Finally, users can calculate mutual assistance capability by using button 3. The capability is calculated automatically. After the calculation, the result is shown by the building unit (see Figure 3).

### 2.4. MAS Sub-Tool

#### 2.4.1. Agent

The model was set to simulate a strong earthquake (intensity 6 (upper) on the Japan Meteorological Agency seismic intensity scale), according to the hazard map announced by the Toyohashi city government [23]. Collapsing buildings, road blockages, and fire spreads were generated. Under the situation, residents evacuated to the designated evacuation site. Therefore, the following agents, as the components of a simplified virtual urban area, were present: building agents, resident agents, point agents, link (road) agents (see Figure 4).

Building agents have certain attributes, such as structure, floor number, and build year. Resident agents are made up of attributes, such as age and gender. Finally, point agents are set at each building agent, and resident agents are generated based on point agents.

#### 2.4.2. Structure of Simulator

The developed simulator includes a road blockage model, a fire spread model, and an evacuation simulator; thus, users can confirm the damage situation of road blockages and fire spread after a huge earthquake. Users can also confirm the evacuation behaviors and mutual assistance activities among residents under the situation.

To enable the simulations’ fixed situations of road blockages and fire spread, a road blockage model and fire spread model were divided from the evacuation simulator. The simulator, including both models was calculated before the evacuation simulator, and the results were then incorporated.

#### 2.4.3. Road Blockage Model

The road blockage model, proposed by Gohnai et al. [24], was incorporated in the developed sub-tool. After setting the probability of building collapse for each building based on structure, floor number, and year of construction, collapsed buildings are generated by using random numbers. When the rubble is spread on a road and then the rest width is under 0.6 m, resident agents cannot pass through the road.

#### 2.4.4. Fire Spread Model

The fire spread model proposed by Ohgai et al. [25] was incorporated as the fire spread model in the developed sub-tool. Fire origins were set by using random numbers. Users could set the wind velocity and wind direction. A fire spread simulation is conducted according to the condition.

#### 2.4.5. Behavior of Resident Agent

Each resident agent is defined as one person. Resident agents evacuated to the designated evacuation site with their families, due to the fact that, in reality, residents will not evacuate without their families. The resident agents performed the following six actions.

(1)Evacuation: each resident agent evacuated from each building to the designated evacuation site. In this model, senior children in elementary school (10 and older) were able to evacuate alone. Children less than 10 years old evacuated with his/her parent.(2)Waiting rescue: the resident agent who was buried under a collapsed building would wait for help. The resident agent who was in a burning building had no support from other resident agents, because in the real world, it is difficult for residents to rescue someone who delays escaping from a fire.(3)Rescue victims: when resident agents discovered a victim in need of help within the perception range (the range that residents can find a person who requires mutual assistance) during evacuation, they took part in the rescue activity. However, resident agents with children less than 10 years of age were given priority in evacuation. When the total expected value of resident agents participating in rescue activities exceeded 1, they could rescue a victim. When the total expected value was not greater than 1 after a lapse of 5 min from earthquake generation, the resident agent gave up rescue and restarted evacuation.(4)Those in need of evacuation support: the residents who could not evacuate individually, such as elderly people and disabled people, waited for other residents’ help.(5)Those supporting evacuation: When resident agents discovered a resident in need of evacuation support within the perception range, they provided evacuation support. However, resident agents with children less than 10 years of age were given priority.(6)Awaiting public support: in the following three situations, resident agents could evacuate, even when performing mutual assistance activities. Therefore, when residents are in the following situations, they should wait on support from public institutions, such as the local fire or rescue teams: (1) a resident agent in a burning building; (2) a resident agent who cannot be rescued by another in situations where the total expected value is not greater than 1; (3) a resident agent who cannot reach the designated evacuation site due to road blockage.

In this simulator, residents do not perform the initial firefighting during evacuation. In addition, it is unlikely that residents, in reality, will start evacuation simultaneously. Therefore, in this study, residents do not start evacuation all at once; rather, they initiate evacuation at random.

#### 2.4.6. Improvement of the Structure

Using the previous fundamental development, it was difficult to incorporate the road blockage model mentioned in Section 2.4.3 and the fire spread model mentioned in Section 2.4.4 correctly. In this sub-tool, the building and the disaster situation (fire spread and rubble generation) were constructed using a grid cell model, and the evacuation route was constructed using a network model. For this, precise evaluation in units of 3 m × 3 m has become possible. The simulator was divided into two models, a fire spread/building collapse model and an evacuation simulation model. This made it possible to perform a fixed simulation of the disaster situation.

#### 2.4.7. Simulation Flow

Figure 5 shows the simulation flow. First, the virtual space is constructed and resident agents are generated. Then, the results of the other simulator, incorporating the road blockage model and the fire spread model of all steps, are passed, before starting residents’ behavior. Then, they are expressed on the virtual space in each step. Residents judge their actions by one second units. One second is 1 step. According to the progress of the step, their results of the other simulator are updated.

The resident agents then examined and judged the behaviors mentioned in Section 2.4.5, under fire spread, building collapse, and road blockage. Basically, the residents judged evacuation and went to the nearest evacuation site. When residents found road blockages, they explored other routes. If they could not find other routes, they waited for public help. Residents who were buried under a collapsed building “judged” weather to wait for rescue (this means “waiting for rescue/support for evacuation”. Resident agents who discovered victims in need of help on their way to evacuation judged whether to go to the victim (this is “go to a victim”). When residents gathered, and the total expected value of resident agents participating in rescue activities exceeded 1, they could rescue a victim.

#### 2.4.8. Interface

Figure 6 shows the interface of the simulator. Using the interface, users can set the following parameters.

(1)Evacuation start time: the time from earthquake generation to starting evacuation.(2)Perception range: the range of residents’ capability to discover individuals who require mutual assistance.(3)Evacuation speed: the walking speed of a resident in evacuation.(4)Ratio of resident agents who require evacuation assistance: the ratio of resident agents who require evacuation assistance for each age group.(5)Mutual assistance setting for children aged 10 to 15: whether or not the setting item performs mutual assistance for children aged 10 to 15 years.(6)Road blockage setting: item to be set to select whether the road is closed by an earthquake or fire.

By changing parameters 1, 2, and 3, at the same time, the situation considered can be changed, according to factors such as weather changes and time of day.

### 2.5. Examination of the Method to Explore a CDMP

Table 2 shows the method to explore a CDMP using the developed tool. The CDMP is made via bottom-up discussions of local residents. Therefore, it is necessary to deepen the importance of mutual assistance among local residents and to enhance awareness for making the CDMP. Therefore, using two sub-tools to understand pre-study WSs seems useful as a first step. In particular, the effects of reducing human damage via mutual assistance, and the necessity of promoting mutual assistance capability are explained through simulator animation. In addition, the introduction of cases involving good mutual assistance activities is conducted. However, the necessary data to use two sub-tools were not collected at that time. Therefore, the previous evaluation result from the other district is used.

After the decision on making a CDMP, a questionnaire survey was conducted to collect data for using the sub-tools. For the examination of CDMP, objective and quantitative evaluation of mutual assistance capability and a grasp of the individuals who cannot evacuate to a designated evacuation point without help, are necessary.

After evaluating capability, workshops were conducted to make the plan.

## 3. Results

### 3.1. Demonstration for Evaluating Revised Tool

#### 3.1.1. Target Area and Parameters

To evaluate and analyze the developed tool mentioned in Section 2, a demonstration in an actual area was conducted. The target area was an elementary school area—Hacchou Elementary School in Toyohashi, Aichi prefecture—including the area with high vulnerability to earthquake disasters (see Figure 7). Using the GIS sub-tool, the area was evaluated to extract the district (neighborhood association area) with low mutual assistance capability.

The extracted area was applied by the MAS sub-tool. For using the MAS sub-tool, the following three cases were set: (1) the presence of mutual assistance activities during evacuation in the morning; (2) the absence of mutual assistance activities in the morning; (3) the presence of mutual assistance activities during evacuation in the evening. The mutual assistance capability was changed according to the time of day caused by the commuting population. The cases in the morning and evening were calculated, and the calculations reflecting the above scenarios were carried out ten times. The base parameters are illustrated in Table 3.

#### 3.1.2. Data for Evaluating the Revised Tool

To verify the revised tool, GIS data of urban area information for operating both sub-tools were obtained from the Japanese National Land Numerical Information download service.

For using the GIS sub-tool, the resident information (male/female and age) by the building unit of the target district is necessary. To make the data, the census population data (small section unit) obtained from the portal site of official statistics of Japan were modified according to the result of the estimation of family composition of each building by field survey. The data mentioned above are used freely by everyone.

For using the MAS sub-tool, the information of the person who cannot evacuate individually and who requires help (including how many in each building) is necessary. To make these data, in actual town planning, a questionnaire survey to get the data, with permission from residents, is necessary. However, the collection of actual data is impossible for a case study. Instead of the data, at first, the number of people needing help was calculated according to the ratio of the people needing help for the whole city. Afterwards, it was modified by a field survey (and after listening to residents).

#### 3.1.3. Analysis of the Results

Figure 8 shows the simulation results in map form. Table 4 shows the result by numerical value reflected in cases 1 to 3. These numerical values are the total value of the 10 simulation results, except for the number of residents. Table 5 shows each result of the revised tool and the previous tool.

Users can change the perception range: in the morning (cases 1 and 2), residents had a wide perception range compared to the evening (case 3). In this demonstration, the perception range in the morning was set at 15 m, and the range in the early evening was set at 9 m.

In addition, the constitution of household members changed: in the morning hours, all household members were in their respective buildings. The total population of 398 was the same as the actual total population. In the evening hours, almost all residents in the virtual space were stay-at-home wives, elderly persons, and students. Commuters were not home as yet. Considering these situations, the total population was assumed as 301 persons.

Comparing case (1) to case (2), (e) the total number of people receiving mutual assistance and (f) mutual assistance success rate, the value of case (1) is higher than case (2). This is the natural outcome when comparing the presence and absence of mutual assistance. However, it is significant to show the effect of mutual assistance activities numerically. If a huge earthquake occurred, 14 residents would be rescued by mutual assistance activities in this area. In this result, users can understand the importance of mutual assistance activities.

Comparing case (1) to case (3), (e) total number of people receiving mutual assistance and (f) mutual assistance success rate of morning and evening, the value in the morning was higher than that of the evening. In this result, it is shown that mutual assistance capability activities by neighborhood changed in accordance with the timeframe. Users can thus understand the need for countermeasures, such as the promotion of mutual assistance activities for timeframes when the capability is low.

In this way, users can easily understand the effect of mutual assistance during resident evacuation. Accordingly, an understanding of the importance of mutual assistance should be promoted.

In addition, by displaying the results in map form, users can easily comprehend the number of persons who were helped by other residents. Moreover, users can perceive the places where residents could not receive rescue or support. Using this information, residents can explore countermeasures to reduce the number of residents who cannot receive rescue or support.

#### 3.1.4. Evaluation of the Tool

To verify the usability and issues of the developed tool mentioned in Section 3.1, two hearing-based surveys were administered on local government staff. An overview of the hearing-based surveys is presented (see Table 6).

One survey was conducted on the staff who knew the situation of the target district. They were suitable for evaluating the validity of the evaluation results of the tools, and the exploration of mutual assistance activities, for improving mutual assistance using the tools, based on careful consideration of the actual conditions of the target area.

Another was conducted on the staff of Kobe city. During the Great Hanshin-Awaji earthquake, many old wooden houses collapsed in Kobe city. However, mutual assistance activities were actively carried out and many lives were saved.

From the experience, Kobe city is actively supporting the creation of disaster mitigation plans on a district level by residents. The mutual assistance activities at the time of disaster occurrence are incorporated into the plan.

The fire department staff are in charge of the disaster mitigation activities of each neighborhood community association, such as disaster drills. The staff support their activities.

From the above, it seems that the staff are suitable at evaluating the validity of the mutual assistance method via a GIS sub-tool, as well as a simulation of mutual assistance behavior by an MAS sub-tool, and usability of the tools for exploring community-based activities for disaster mitigation, including mutual assistance activities, based on careful consideration of the experience-related mutual assistance activities mentioned above.

The hearing-based surveys method is as follows. At first, by explaining the outline of the tool and the usage detail (as mentioned in Section 3, Section 4 and Section 5), an understanding of the evaluation method of mutual assistance capability, as well as the outline, function, and usability regarding the developed tool, was promoted to participants. After that, participants answered each item.

Comments from the participants were then obtained (see Table 7).

Regarding the GIS sub-tool, the following usability characteristics were obtained. Extracting the area showing low capability was simplified by calculating wide range. It is easy for residents to understand the area showing low capability. Considering obtained opinions, it seems that visualizing mutual assistance capability, quantitatively, was evaluated. For this, local governments can approach the neighborhood association of the area for enhancing the improvement of low capability. Getting the consensus-building of the neighborhood association seems to become easy by using this evaluation result.

Regarding the MAS sub-tool, the following usability characteristics were obtained.

The effect of mutual assistance to reduce human damage is visually shown. It is easy for residents to understand the effect of mutual assistance. Therefore, understanding the importance of mutual assistance should be promoted. In addition, the awareness that it is important to know the residents who cannot evacuate individually (such as elderly people) in advance is promoted. This tool can simulate a reflection of opinions obtained by exploring the contents of community-based activities. Considering obtained opinions, it seems that comparing the simulation results, with or without mutual assistance, eases understanding of the effect of mutual assistance. In addition, the function to change parameters, such as perception range, seems useful for exploring the contents of community-based activities, including mutual assistance.

Evaluating the validity of the mutual assistance method via the GIS sub-tool as well as a simulation of mutual assistance behavior by the MAS sub-tool, was obtained from the fire department staff of Kobe city. However, some unexpected damage caused by an earthquake may occur (such as the Great East Japan earthquake). Various situations of damage, such as building collapse and road blockage. are conceivable. Therefore, it will not necessarily correspond to the simulation results. It is necessary to explain this point to residents sufficiently.

In addition, the difficulty of data collection was identified. This evaluation method of mutual assistance capability requires detailed personal data, such as age, gender, and household members. Thus, it is necessary to get an agreement from residents (in regards to the developed tool) in real areas.

### 3.2. Evaluating the Method to Explore a CDMP

#### 3.2.1. Outline of the Target Area

The target area was Ushihachi district in Toyokawa city, Aichi prefecture, Japan. Six densely-built up districts with a high risk of damage (from a major earthquake) were selected by the Toyokawa city government. One of them was Ushikubo wide district. The wide district has a plan for promoting improvement, including widening road widths and rebuilding. Ushihachi district is a district that belongs to the Ushikubo wide district. The target area is divided into 15 small groups (Figure 9). The awareness of disaster mitigation is high. In addition, the community-based activities for disaster mitigation have been conducted.

#### 3.2.2. Pre-Study Workshop

For the first WS, the advanced cases for promoting mutual assistance capability were collected and organized for information sharing, to examine CDMP. Based on the result, the activities for promoting mutual assistance capability were explained to the members for disaster mitigation of the neighborhood association. In addition, the outline of the two sub-tools the authors developed was explained for understanding the necessity for making CDMP. The GIS sub-tool was used for understanding the image of the area with low capability. The MAS sub-tool was used for understanding effects of mutual assistance activities. The evaluation result of the other district mentioned in Section 3.1 was used.

As a result, the decision for making CDMP from the neighborhood association was obtained. In addition, concerning using the developed tool, the following opinions were obtained.

Using a GIS sub-tool—there is no problem making it.In regards to the MAS sub-tool, the simulation result seems to include privacy information. Due to the concerns, obtaining consensus among residents of the neighborhood association is necessary before using it.

Considering these opinions, using the MAS sub-tool was extended after obtaining consensus.

#### 3.2.3. Questionnaire for Data Collection

The questionnaire to collect the necessary data for using the developed tool was conducted. The items of the questionnaire are shown in Table 8. The questionnaires for the households belonging to the neighborhood association were distributed and collected by the members of the association. The questionnaires for the households that did not belong to the association were put in a post box and returned by mail. The response ratio was 97% and 14%.

Based on the data, the evaluation of mutual assistance capability was conducted. In Section 2.2, the outline of the GIS sub-tool was explained. In the explanation, after evaluating the expected value of each household and weights by distance, capability is evaluated by building unit.

However, the target area is divided into 15 small groups and the connection of the community among each small group is strong. It is thought that there is no limitation due to the distance for conducting mutual assistance activities after a huge earthquake. Therefore, the evaluation of the rescue expectation value was conducted by the small group unit. The weighting by distance was not conducted. The evaluation means mutual assistance capability. The evaluation result is shown in Figure 10 and Figure 11, and Table 9.

#### 3.2.4. Examination of the Draft of CDMP

The second WS was conducted to understand the capability of current conditions of the target area and explore improvement. The evaluation result of the mutual assistance map was explained by the WS staff (authors’ lab members) using the materials mentioned in Section 3.2.4. The details are as follows.

In case 1 (Table 9, item 4), there is a shortage concerning the mutual assistance capabilities of 11 small groups. The total capability of 15 small groups is −16.821, because there are many people who need assistance.

After that, a discussion was conducted to explore the improvement method. According to the opinions from residents who took part in the first WS, the main content of the CDMP is evacuation and a support method with mutual assistance, for the initial response right after a huge earthquake. Some opinions, as shown below, were obtained from the participants, after confirming the analysis result of the mutual assistance map.

The definition of the person who needs help seems to be too wide. For example, people under 10 years old should exempt from the number of people who need assistance. If it is an infant, it can be carried by one adult. If the individual is older than an infant who can walk, then one adult can hold hands with the individual and guide him/her to evacuate.Many young households do not belong to the neighborhood association. If they participate in the association’s activity, it will lead to the improvement of mutual assistance capability.There are points of consideration to explore concerning the support method used for the individual in need of help. One is the high aging rate of the whole district. The other is that the number of households, by the small group unit, is largely different.It is necessary to build a mutual assistance system by block unit (where the number of households is equally divided), instead of the current small group unit.A cooperation system between the group with low capability and the group with high capability is required.There is a shortage concerning the mutual assistance capabilities of the whole Ushihachi district. Thus, a cooperation system with other districts surrounding the Ushihachi district is also required.

#### 3.2.5. Revision of the Draft of CDMP

In the third WS, at first, based on the opinions in the second WS, case 2 (Table 6, item 5), mutual assistance capability without people under 10 years old from the number of people who need assistance was calculated. The result was compared with the previous result of case 1 (See Figure 12).

In case 2, there is a shortage concerning mutual assistance capability of nine small groups. In the total capability of 15 small groups is −2.821. In comparison to case 1, the shortage value becomes low, because the number of people who need assistance is reduced.

After that, the revised points were discussed by participants. Finally, a CDMP was created (see Figure 13).

## 4. Discussion

### Questionnaire Survey to Residents

The two questionnaires were conducted to verify the usability and issues for creating a CDMP as mentioned in Section 3.2. One is for grasping the effect of the tools to promote the awareness of mutual assistance and creation of CDMP. The other is for grasping the effect of the mutual assistance map to promote detailed planning. The outline of each questionnaire is shown in Table 10. The results are shown in Table 11.

In regards to the first questionnaire, through Questions 1 to 4, all participants answered 1 or 2. Although there may be a possibility that the participants may have had a high interest in disaster mitigation (they were members of the disaster mitigation team in the neighborhood association), the evaluation of the tools from the participants seemed high. In particular, the (easy to understand) mutual assistance capability (by the mutual assistance map) and the effect of promoting awareness of the necessity of mutual assistance activities, by the simulator from the detailed comments in free description, was expected.

In the second questionnaire, Question 1, all participants answered 1 or 2. The grasp of the person who needs assistance is useful for enhancing mutual assistance capability. Concerning Question 2 (Q2), although the ratio of the number of people who answered 1 or 2 was high; two persons answered 3.

In regards to Q3 and Q4, to evaluate the mutual assistance map, the ratio of the number of people who answered 1 or 2 was high; two persons answered 3.

In regards to Q5, to evaluate the method to explore a CDMP, such as this time, the ratio of the number of people who answered 1 or 2 was high; two persons answered 3.

Concerning the validity of developed tools, from the results of the first questionnaire, it is thought that the utilization of both techniques is effective, as explored in Section 2.3. In particular, grasping the concept of mutual assistance capability, by the mutual assistance map, and enhancing awareness of the necessity of mutual assistance activities by both techniques, seems to be useful.

As a detailed opinion, “it is easy to understand the area with high or low human damage by using the evacuation simulator” is shown as useful. On the other hand, the concern about the collection of detailed personal information was shown. This concern seems to influence the decrease of answer “1” of question 2 in the first questionnaire. Considering this point, using the simulation result of other districts as reference material is effective at the step for enhancing awareness. The utilization of the simulation tool with the data of the target area is suitable at the step for exploring detailed mutual assistance activities, after obtaining agreement among residents.

In addition, extracting detailed opinions, identifying issues, and encouraging discussions to improve these issues (to create a CDMP based on the mutual assistance map) is shown.

Concerning the validity of the proposed method, from the results of the second questionnaire, usefulness of the grasp of the person who needs someone’s support, and exploration of support contents for the person, were shown. In addition, utilization of the mutual assistance map is useful. Although the ratio of the number of people who answered 1 and 2 was low compared with the first questionnaire, it seems to be useful enough. The impact was high because it was the first time for the participants to know the tool in the first questionnaire. In the second questionnaire after using the tool, the impact is generally reduced. In other words, this is a more accurate evaluation after participants understand the tool.

Considering the above, the exploration method of the CDMP suggested in this study is useful for promoting mutual assistance capabilities.

In this way, a methodology of WSs for making CDMPs by using the developed support tool is useful. At first, in the pre-study WS, understanding the tool was promoted. For this, participants can understand the mean of the evaluation results by the tool. Awareness for making CDMP among residents was also promoted.

In the WS for making CDMP, evaluation results were provided by numerical value and map form. These quantitative materials promoted concrete exploration, such as targeting the person in need of assistance and a cooperation system.

The necessary data for evaluation was collected via a questionnaire in a small group unit. This questionnaire become an opportunity to get a good understanding of the people who needed assistance.

On the other hand, anxiety was expressed, such as data leaks, misuse about the collection of detailed personal information, and detailed simulation results showing individuals who cannot evacuate without help. This point seems to be demonstrated in the decrease in the answer “1” of Q2. Considering this point, using the simulation results of other districts as reference material is effective in the step for enhancing awareness. The utilization of the simulation tool with the data of the target area is suitable at the step for exploring detailed mutual assistance activities, after obtaining agreement among residents. In addition, meticulous attention when managing data and handling simulation results, is required.

## 5. Conclusions

This paper developed a methodology of WSs for making CDMPs, by using the revised support tool to explore mutual assistance activities, based on the results of our previous study.

In addition, the usabilities and issues of the method for exploring CDMP are clarified through demonstration and examination of CDMP in an actual district. The usability of the method was shown. Specifically, it is effective at revising CDMPs, and for raising resident awareness on the importance of mutual assistance.

In addition, the effectiveness of the tool that supports these effects was confirmed.

By utilizing the GIS sub-tool—evaluating mutual assistance capabilities for wide areas, and extracting the areas showing low capability—was simplified. Local governments and experts can extract the areas showing low capability.

By providing numerical figures, it becomes possible for local governments and experts to explain the necessity of improving mutual assistance activities to neighborhood associations in districts with low capability. In addition, by utilizing the MAS sub-tool, it becomes possible to show the reduction of human damage through mutual assistance activities. The information encourages resident awareness concerning the necessity of promoting mutual assistance capabilities. Through these effects, obtaining agreements to start exploration (to improve capability) from residents of extracted areas, is encouraged.

In addition, it is important to know the residents who cannot evacuate individually, such as elderly people, in advance.

In this way, it was shown that the developed method can support the exploration of CDMP and enhance mutual assistance. These findings can improve the situation (i.e., the lack of technology to promote mutual assistance capability) among residents through making CDMPs.

In future works, the evaluations—before vs. after using the method through actual community-based activities—should be compared. In addition, it is necessary to consider a data collection and management method so residents do not feel anxiety over their information being collected.

Finally, in regards to the MAS sub-tool, the need to address issues pertaining to relational uncertainty and probabilistic relational modeling, such as in [29], is pointed out.

## Figures and Tables

**Figure 1 ijerph-18-03814-f001:**
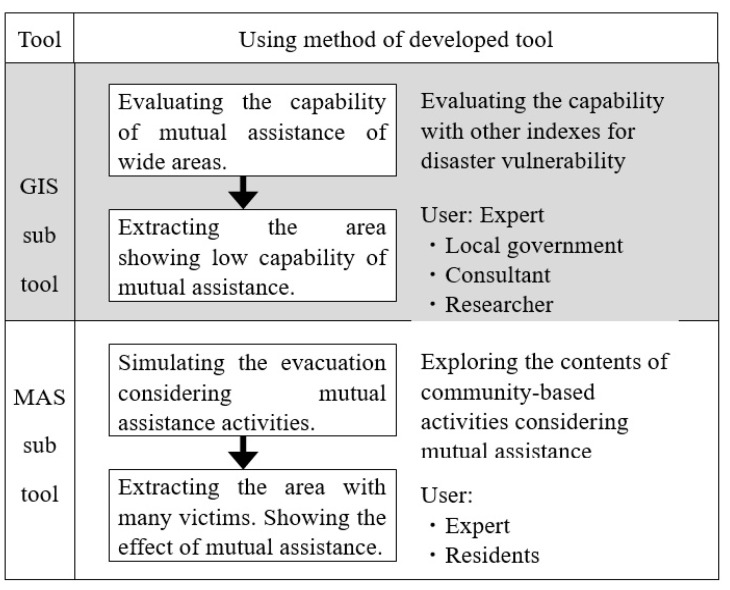
The process for promoting mutual assistance capability. Abbreviations: GIS = geographic information system; MAS = mutual assistance activities.

**Figure 2 ijerph-18-03814-f002:**
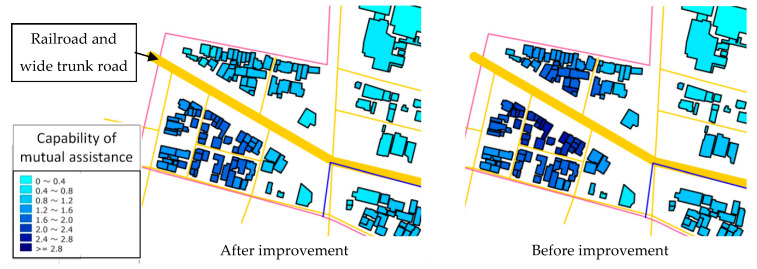
Example of evaluation results before and after improvement.

**Figure 3 ijerph-18-03814-f003:**
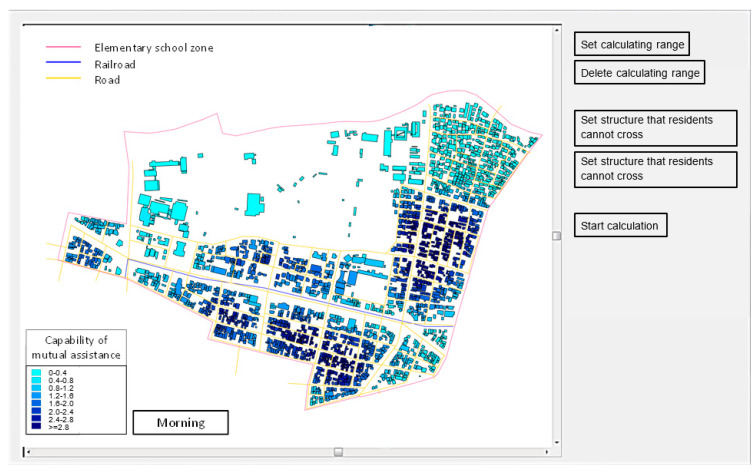
The GIS sub-tool interface.

**Figure 4 ijerph-18-03814-f004:**
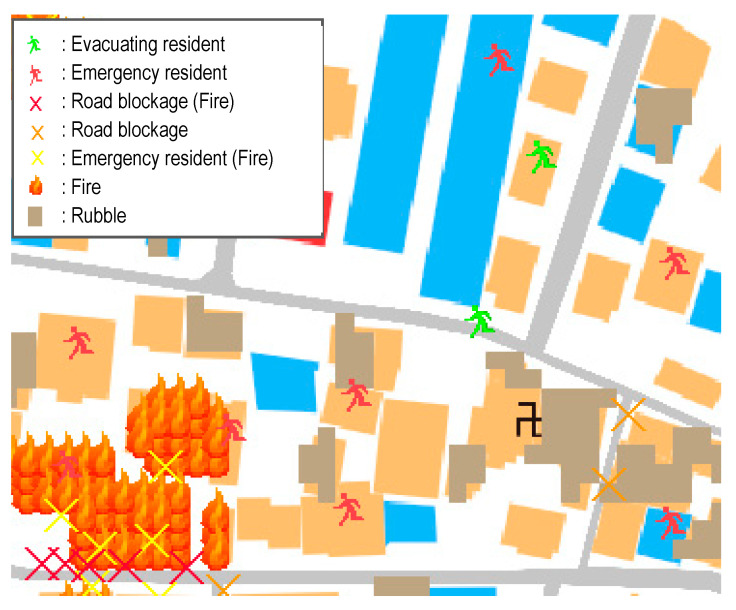
Example of the multi-agent system (MAS) sub-tool.

**Figure 5 ijerph-18-03814-f005:**
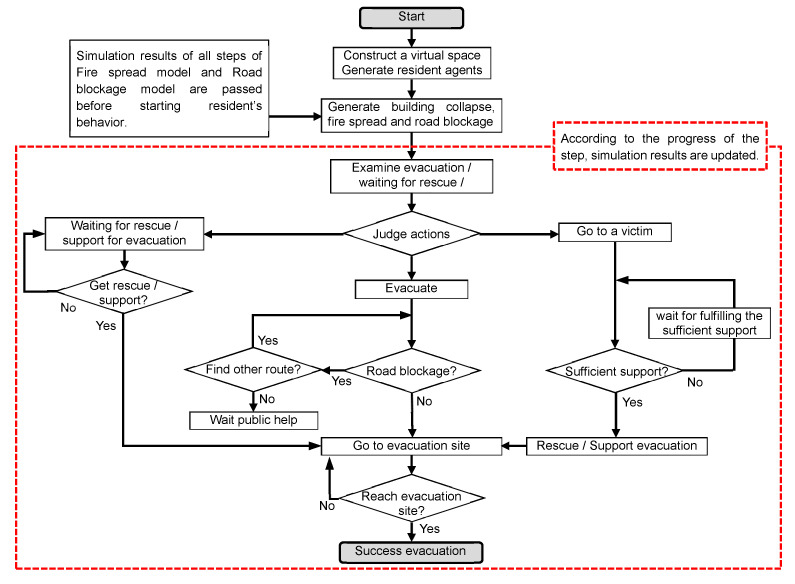
The flow of the MAS sub-tool.

**Figure 6 ijerph-18-03814-f006:**
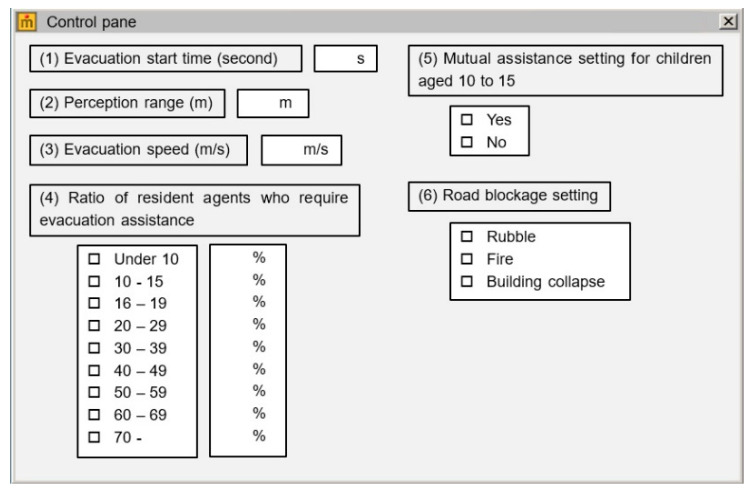
The control pane of the MAS sub-tool.

**Figure 7 ijerph-18-03814-f007:**
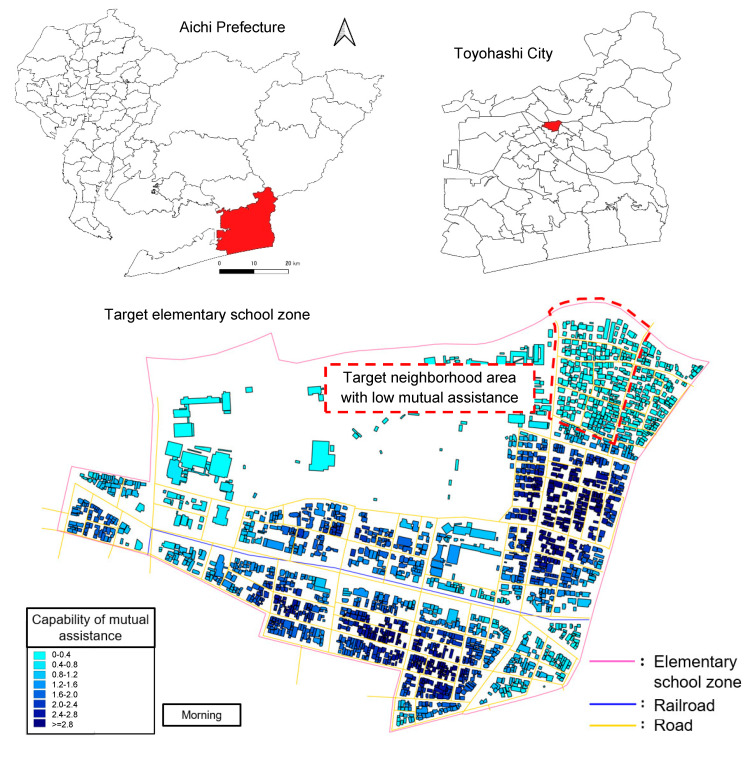
The target area for demonstration.

**Figure 8 ijerph-18-03814-f008:**
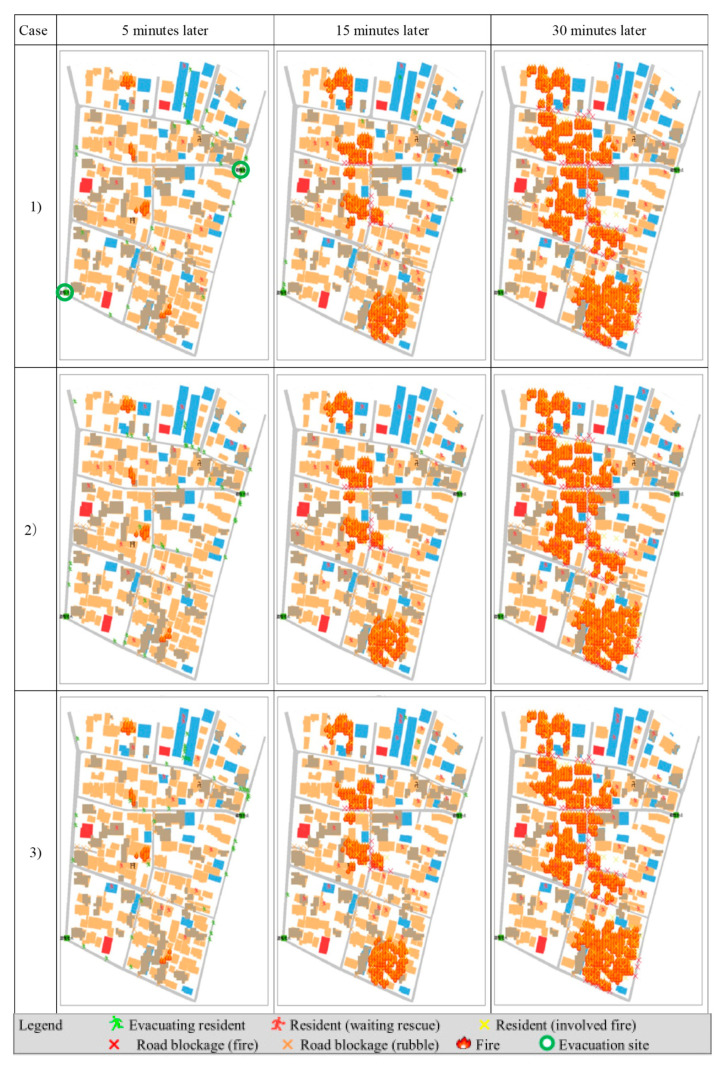
The simulation results in map form.

**Figure 9 ijerph-18-03814-f009:**
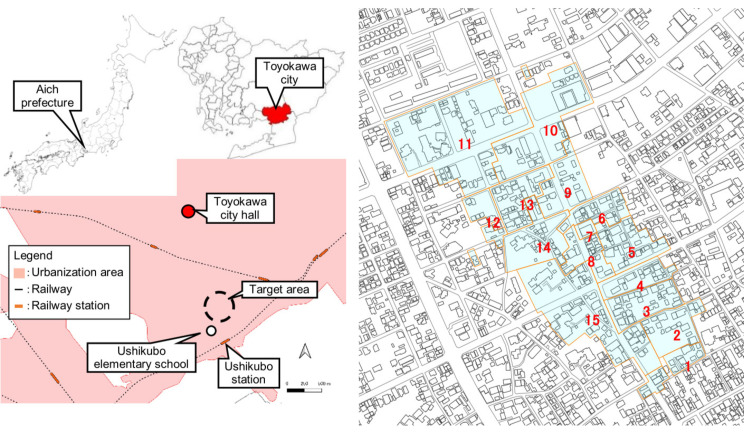
Target area and 15 small groups.

**Figure 10 ijerph-18-03814-f010:**
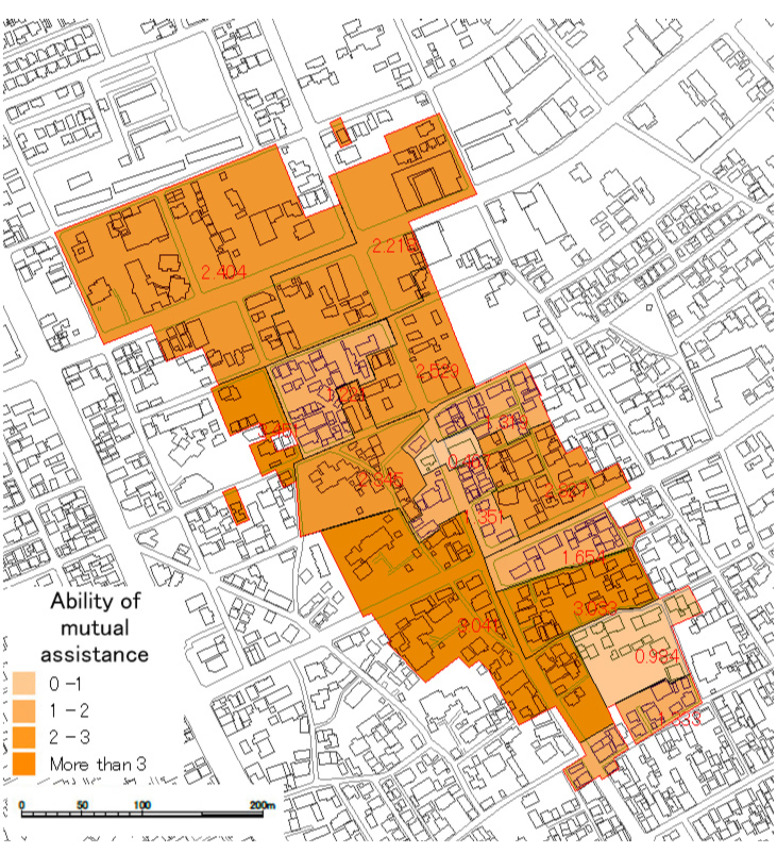
Evaluation result of the capability.

**Figure 11 ijerph-18-03814-f011:**
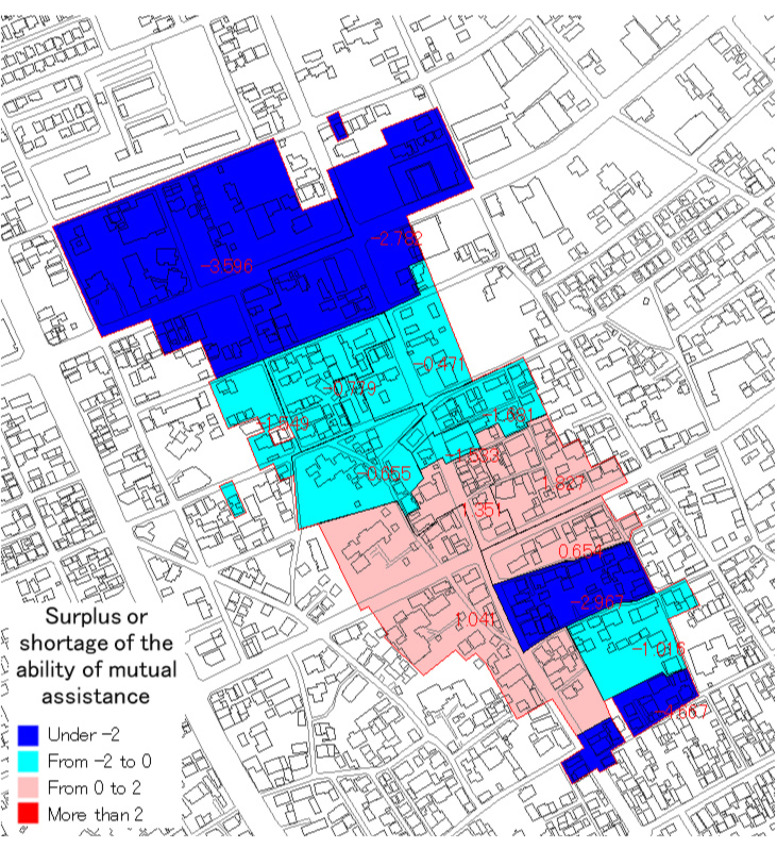
Surplus or shortage of the capability.

**Figure 12 ijerph-18-03814-f012:**
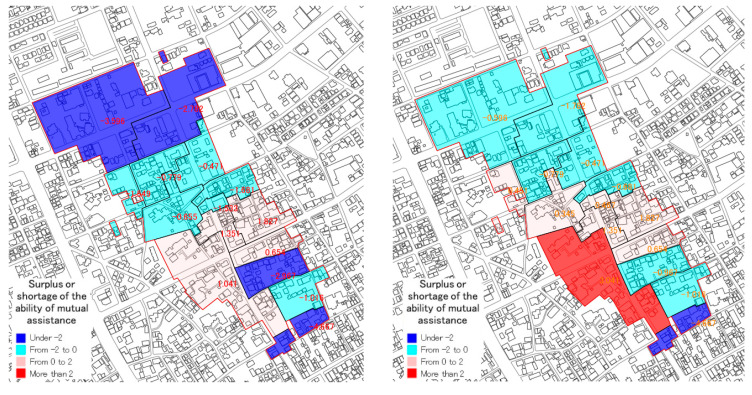
Comparison between case 1 and case 2 (surplus or shortage of mutual assistance capability).

**Figure 13 ijerph-18-03814-f013:**
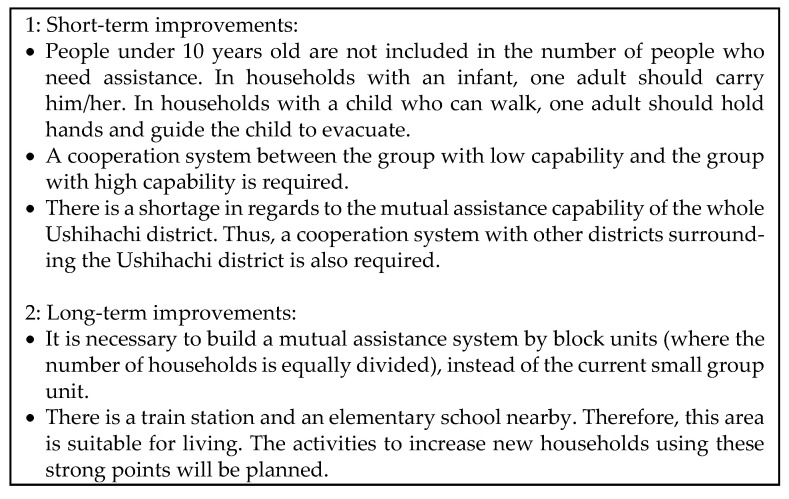
The contents of the CDMP for promoting mutual assistance capability.

**Table 1 ijerph-18-03814-t001:** The expected value in accordance with age and gender.

Age	Men’s Strength	Women’s Strength	Executing Rate	Men’s activity RATE	Women’s Activity Rate	Men’s Expected Value	Women’s Expected Value
10	1	0.85	0.228	0.76	0.24	0.1733	0.0465
20	1	0.76	0.228	0.76	0.24	0.1733	0.0416
30	0.96	0.76	0.229	0.72	0.28	0.1583	0.0487
40	0.93	0.73	0.298	0.72	0.28	0.1995	0.0609
50	0.9	0.72	0.228	0.63	0.37	0.1293	0.0607
60	0.84	0.7	0.191	0.74	0.26	0.1187	0.0348
70-	0.78	0.65	0.129	0.75	0.25	0.0755	0.021

**Table 2 ijerph-18-03814-t002:** The method to explore a CDMP using the developed tool.

Step	Role of the Tools	Effect	Target
Step 1: pre-study workshop. Understanding the necessity for making CDMP.	A: Understanding the area with low capability. B: Understanding effect of mutual assistance activities.	Promoting awareness for making CDMP. Understanding the necessity of the data for calculating mutual assistance capability.	Tool user: university consultant municipality Participants: residents
Step 2: questionnaire survey. Collecting the data for calculating the mutual assistance capability	A: Evaluating mutual assistance capability.	Understanding the area with low capability.	Tool user: university consultant municipality
Step 3: Workshop. Understanding capability in the current condition, of the target area, and exploring improvement.	A: Understanding the area with low capability. B: Understanding the place—of the household—with the residents who cannot receive help from neighbors.	Promoting awareness to improve the issues. Understanding the issues surrounding mutual assistance capability.	Tool user: university consultant municipality. Participants: residents.
Step 4: Workshop. Exploring the countermeasures to reduce the residents who cannot receive help from neighbors.	A and B: Understanding the effect of the improvement.	Understanding the difference of effect for each case. Promote exploration.	Same as Step 3.

A: GIS sub-tool; B: MAS sub-tool.

**Table 3 ijerph-18-03814-t003:** Base parameters.

Parameters	Case 1	Case 2	Case 3
Presence or absence of mutual assistance	presence	Absence	presence
Change in time	morning	morning	evening
Wind speed	5 m/s	same	same
Wind direction	northwest	same	same
Evacuation speed [26]	1.5 m/s	same	1.0 m/s
Perception range [27] (*)	15 m (5 cells)	0 m (0 cell)	9 m (3cells)
Ratio of residents who could not evacuate [28] (**)	60–69: 0.1 70 & over: 0.2 other: 0.05	same	same
Road obstruction by rubble (***)	passable	same	same
Road obstruction by fire	impassable	same	same
Agent can be out of the building at the time of the fire	no	same	same

(*) Users can set the value by 3 m unit (1 cell) based on the characteristic of the target area. In the base parameter, the values were set based on Reference [27]. (**) Users can set the value based on the population characteristic of the target area. In this case, the ratio was set based on Reference [28]. (***) Passable means residents can evacuate the space between rubbles on the road. Impassable means residents cannot evacuate by fire spread.

**Table 4 ijerph-18-03814-t004:** The numerical value of simulation result.

Case	(a) Number of Residents	(b) (*) Total Number of Successful Evacuations	(c) Evacuation Success Rate (%)	(d) Number of People Requiring Mutual Assistance	(e) Total People Receiving Mutual Assistance	(f) Mutual Assistance Success Rate (%)
(1)	398	2917	73.25	1085	148	13.58
(2)	398	2787	70.05	1105	0	0.00
(3)	301	2172	72.16	802	44	5.36

(b), (d) and (e) are total number of the 10 simulation results; (c) was calculated as follows. At first, (b)/10 = average of successful evacuation; next, the average/(a) = (c); (f) this was calculated as follows. (e)/(d) = (f); (*) the number of successful evacuations of each simulation is below.

**Table 5 ijerph-18-03814-t005:** Each result of the revised tool and the previous tool.

	1	2	3	4	5	6	7	8	9	10	Total	Average
A	293	296	287	290	289	295	293	296	291	287	2917	291.7
B	258	304	289	306	321	313	271	297	261	295	2915	291.5

A: 10 simulation results of revised tool; B: 10 simulation results of previous tool.

**Table 6 ijerph-18-03814-t006:** Outline of hearing surveys.

**First Hearing-Based Survey: A**	
Date	18 January 2016
Participants	Disaster mitigation division staff of Toyohashi City Urban planning division staff of Toyokawa City
Items	Usability of the developed tool Issues of the tool Necessary functions
**Second Hearing-Based Survey: B**	
Date	19 January 2017
Participants	Fire department staff of Kobe City
Items	Validity of evaluation method Usability of the developed tool Issues of the tool Necessary functions

**Table 7 ijerph-18-03814-t007:** The comments obtained by the hearing surveys.

Evaluation	Comments
Usability	GIS sub-tool Calculating wide range was sufficient. Extracting the area with low capability was simplified. (A) This tool can display factories and offices where many people work (showing high mutual assistance). (A) The mutual assistance capability is visually shown. It is easy for residents to understand the areas showing low capability. (A, B) Local governments can approach the neighborhood association of the area for enhancing the improvement of low capability. Getting consensus-building from the neighborhood association becomes easy. (A)
	MAS sub-tool The mutual assistance capability is visually shown. It is easy for residents to understand the effect of mutual assistance. (A) This tool can simulate a reflection of opinions obtained by exploring the contents of community-based activities. (A) Understanding the importance of mutual assistance may be promoted. (A) The awareness that it is important to know the residents who cannot evacuate individually (such as elderly people) in advance is improved. (A) As an example of using the method, it may be better to see the simulation once without explanation first, in order to get residents’ interest. Next, a facilitator explains the simulation. After that, the residents see the simulation again. (B) When examining the activities of mutual assistance among residents, the simulation result is effective for them to have an idea of mutual assistance. Showing the simulation result of mutual assistance has a strong impact. Therefore, residents can have interest. (B) The confirmation of simulation can trigger discussion. (B)
Validity	GIS sub-tool The evaluation method of mutual assistance is reasonable compared to the real activities of mutual assistance during the Hanshin-Awaji earthquake in 1995. However, various damage situations are conceivable. Therefore, it will not necessarily follow the simulation results. It is necessary to explain this point to residents sufficiently. (B)
	MAS sub-tool As with the GIS sub-tool, the behavior of residents (agents) for mutual assistance is reasonable compared with the real activities of mutual assistance during the Hanshin-Awaji earthquake. (B)
Issues	When explaining the results of the tool to residents, clearer explanations will be required. (A) Data collection is difficult. (A,B)

A and B are the targets of the hearing survey mentioned in Table 7.

**Table 8 ijerph-18-03814-t008:** Items on the Questionnaire.

Q1: Number of persons in your household and the constitution.
Q2: Are there any persons who cannot evacuate and need someone’s help in your household?
Q3: Do you have enough persons to support him/her in your household? (If you answer “No” at Q2, please skip this.)
Q4: Are there any persons who cannot evacuate and need someone’s help in your neighborhood (exclude your household)?
Q5: Are there any persons who can support the person mentioned above (Q4) in your household? (If you answer “No” at Q4, please skip this.)
Q6: Which distance can you support the person mentioned at Q4?
Q7: If possible, could you mark your house in the below map?
Q8: Could you tell us any opinion about the answers mentioned above, if you feel?

**Table 9 ijerph-18-03814-t009:** The numerical values of the results of the questionnaire and evaluation.

Items	The Number of Small Groups															Total
	1	2	3	4	5	6	7	8	9	10	11	12	13	14	15	
(1): People who need assistance	6	2	6	1	1	3	2	0	3	5	6	5	2	3	2	47
(2): Pople under 10 years old	0	0	2	0	0	1	2	0	0	1	3	2	0	1	2	14
(3): Ability of mutual assistance	1.3327	0.9839	3.0331	1.6542	2.8272	1.3194	0.4674	1.3514	2.5286	2.2181	2.4041	3.451	1.2214	2.3447	3.0414	30.179
(4): Surplus or shortage by (3) – (1)	−4.6673	−1.0161	−2.9669	0.6542	1.8272	−1.6806	−1.5326	1.3514	−0.4714	−2.7819	−3.5959	−1.549	−0.7786	−0.6553	1.0414	−16.821
(5): Surplus or shortage by (3) – (2)	−4.6673	−1.0161	−0.9669	0.6542	1.8272	−0.6806	0.4674	1.3514	−0.4714	−1.7819	−0.5959	0.451	−0.7786	0.3447	3.0414	−2.8214

(1) People who are difficult to evacuate when a huge earthquake occurs, such as the elderly, disabled people, infants, pregnant women, foreigners without Japanese language knowledge. (2) The value subtracting the number of people under 10 years old from the number of people who need assistance. In the questionnaire, a person under 10 years old was regarded as a person in need of assistance. However, if it is an infant, it can be carried by one adult. If the individual is older than an infant who can walk, it is thought that one adult can hold the individual’s hand and guide him/her to evacuate. Therefore, the number of people under 10 years old were investigated in the questionnaire. (3) If the value is more than 1, a person can receive enough assistance from neighbors. If the value is more than 2, two people can receive enough assistance. (4) This is the value subtracting the number of people who need assistance from the rescue expectation value. The blue letter means shortage. The red letter means surplus. (5) This is the value subtracting the number of people who need assistance from the rescue expectation value according to the consideration of (2).

**Table 10 ijerph-18-03814-t010:** Outline of the two questionnaires.

Date	Items
First survey 20 January, 2018	For grasping the effect of the tools to promote the awareness of mutual assistance. Participants: 17 members
Second survey 20 March, 2019	For grasping the effect of the mutual assistance map to promote detailed planning. Participants: 14 members

**Table 11 ijerph-18-03814-t011:** The result of the questionnaire.

	Q	Question Items	1	2	3	4	Total
1st	1	Do you think it is easy to understand mutual assistance capability by using the mutual assistance map?	6 (35%)	11 (65%)	0 (0%)	0 (0%)	17
	2	Do you think it is easy to understand the area with high or low human damage by using the evacuation simulator?	3 (18%)	14 (82%)	0 (0%)	0 (0%)	17
	3	Do you think that the two tools are useful in promoting the exploration of the contents of mutual assistance activities?	4 (24%)	13 (76%)	0 (0%)	0 (0%)	17
	4	Do you think that the two tools are useful for enhancing the awareness of the necessity of mutual assistance activities?	6 (35%)	11 (65%)	0 (0%)	0 (0%)	17
	5	Free description	-	-	-	-	-
		It is easy to understand the area with human damage and the effect of mutual assistance by using the evacuation simulator. Promotion of awareness about the necessity of mutual assistance is expected. There is a little fear about the validity of collecting detailed personal information. This information can promote the exploration of concrete discussions. The simulation result of the MAS sub-tool shows the person who cannot escape without help from neighbors. However, this information seems to be privacy-related information. Is this detailed information necessary to explore a CDMP? It will be used after obtaining an agreement and high awareness of the necessity to collect the data.	-	-	-	-	-
2nd	1	Do you think that understanding the person who needs someone’s support is useful for enhancing mutual assistance capability?	6 (43%)	8 (57%)	0 (0%)	0 (0%)	14
	2	Do you think that the exploration of support contents for the person who needs someone’s support is useful for enhancing mutual assistance capability?	5 (36%)	7 (50%)	2 (14%)	0 (0%)	14
	3	Do you think that the mutual assistance map is useful for grasping mutual assistance capability?	5 (36%)	7 (50%)	2 (14%)	0 (0%)	14
	4	Do you think that the mutual assistance map is useful for the exploration of support contents for the person who needs someone’s support?	4 (29%)	8 (57%)	2 (14%)	0 (0%)	14
	5	Do you think that the exploration method of the district disaster mitigation plan is useful for promoting mutual assistance capability?	3 (22%)	9 (64%)	2 (14%)	0 (0%)	14

Answer 1: extremely useful 2: useful 3: not very useful 4: not at all useful.

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
