# Peer review of "A Methodology of Workshops to Explore Mutual Assistance Activities for Earthquake Disaster Mitigation"

_ijerph, 2021, doi:10.3390/ijerph18073814_

Round 1
Reviewer 1 Report
Please, see my report attached.

Author Response
Dear reviewer 1
Thank you for your kind suggestion. I revised my manuscript as attached file.

Reviewer 2 Report
The manuscript "A Tool to Explore Mutual Assistance Activities for Earthquake Disaster Mitigation" by Kazuki Karashima, and Akira Ohgai is very significant about the proposal of the Authors and merits to be published. It is of high interest to the community of risk reduction.
However, a major revision is needed to obtain a standard quality publication.
First, Figures 2, 3, 4, and 6 must be included to consider the manuscript for further revision.
This work seems to lack an adequate scientific treatment in the sense of a quantitative evaluation of the surveys, and a low number of participants. Although, at the state of the art the proposals of the authors remain significant for the treated topic.
More indispensable is to clarify some points which are listed below with an English revision.
Specific requirements:
- lines 43-44, end lines 48, 55, 58, and 61, references are required at the end of the sentences;
- line 83, declare WS "Workshop (WS)";
- rewrite the sentence 87-88;
- line 89, briefly describe the results of [2];
- line 94, is this your conclusion? if no, put a reference;
- line 132, put ":" and lines 133-139 use a), b), c),.. or i), ii), iii),... or rewrite lines 13-139;
- lines 139-140, delete;
- line 141-144, rewrite better;
- line 145-147, rewrite better;
- line 154-163, rewrite better;
- lines 216, 223, and 231 cite Figure 2, 3, and 6 which are not included;
- from line 226 onwards, use strong earthquake as huge earthquake is M>=9;
- line 315 cite Figure 6 that is not included;
- from line 352 onwards, use M/F instead of gender which could be misleading;
- put all the "m" with space, es: 9 m, 15 m...;
- lines 420-422, advances? please explain and rewrite;
- lines 454-459 rewrite better;
- line 467, starts with a sentence;
- lines 468, 471, 477, 480, and 482, modify : : : ... and "after that", "after that" ..., please use a native English speaker corrector;
- lines 613-614, rewrite better;
- lines 660-667, easy easier, easy easier..., important, ... please explain and describe.
Author Response
Dear reviewer 2
Thank you for your kind suggestion. I revised my manuscript as attached file.

Reviewer 3 Report
-
Review “A Tool to Explore Mutual Assistance Activities for Earthquake 2 Disaster Mitigation”
Round 1
- Introduction-Literature review
- As to the initial statements (lines 24-60), I suggest the authors that they support them by including some references of works sharing their points of view.
- Gap identified: “However, there is no method to evaluate the ability of mutual assistance at the district level, identify issues and encourage discussions to improve the ability of mutual assistance.; There is no method to evaluate the ability of mutual assistance at district level quantitatively, identify issues and encourage discussions to improve those issues for creating a CDMP.”. ¿Is there any qualitative method to evaluate mutual assistance?. It is very questionable since the authors already developed such a method. They must include their own studies, pointing out the lacks in their methodology a the gap to be bridged.
- Contribution: “This study aims to develop a support tool to explore and promote mutual assistance activities for district level.”. It is very questionable. As in the case of the gap, they should redirect the paper’s aiming towards the lacks on their own methodology: “Therefore, although the tool development of this study is based on Karashima and Ohgai [14] [15], it is unique because it attempts the revision of a technique for exploring community-based activities considering mutual assistance improving the issues mentioned above.”. However, the novelty of their technique is also questionable since it seems that the reviewed paper just reproduce the methodology presented by the authors in former work already published. Therefore, the authors must strive in clearly showing the advances of the paper they are submitting for publication.
- Methodology
GIS sub tool/Evaluation method:
2.3.1. Mutual assistance evaluation method: For the evaluation for calculating the capability of mutual assistance, the authors refer to the methodology proposed by Akiyama et alt in their work “An evaluation method for the capability of initial response to huge earthquakes and a proposal for the policy of disaster mitigation in Japan”. This work, however, has not been published in any recognized scientific journal, nor could be found in popular scientific repositories such as google scholar, or accessed via internet through common search engines such as Google. It was therefore impossible for this reviewer to evaluate the validity of the assessment methodology employed by the authors.
2.3.2. Improvement to consider the features that are difficult to cross: As to the improvements that the authors have introduced over their former work, which is intended to be the contribution presented in the reviewed paper, the explanations on this features are referred to figure 2, but this figure does not exist. Therefore, as in the case of the evaluation method, the paper does not afford the evaluating the validity of the authors’ methodology.
The reviewed paper does not provide the reader with information enough for the reviewing of the proposed methodology, which makes this work, on its current form, to be clearly below the standards of high-quality scientific publications such as the International Journal of Environmental Research and Public Health. On the other hand, the fact that the methodology is be based on unpublished work contributes to raising serious concerns as to the validity of the methodology embodied in the reviewed paper.
MAS sub tool:
Evacuation model
- Behaviour of residents: probabilities of waiting for rescue/evacuate/going to a victim in judgment?
- Simulation flow:
- step unit: although the road blockage and fire spread models provide an initial, static scenario, the evacuation is a dynamic process. How does the authors have addressed the dynamics of the evacuation flow?. Have the authors employed any system dynamics technique, or discrete event simulation method, to address this issue?
- What does ”wait for next step” mean?. Seems that the flow is broken at this step
- Results
3.1.1. Target area and parameters
What studies support the values of:
- Perception range
- Evacuation speed
- Ratio of residents requiring assistance
- Road blockage setting
3.1.3. Analysis of the results
In this sub-section, the authors show the usefulness of their tool in illustrating the importance of mutual-assistance. However, there are some results that would require further explanation. For example, how the fact that there are more successful evacuations than residents (Table 3, case 1, 398 residents and 2917 successful evacuations) should be interpreted?.
On the other hand, the number of simulations performed (10 simulations) would be too low to confer reliability enough to the results. ¿What is the convergence criterion considered by the authors in the selection of the number of simulations?
Finally, in order to improve the reliability of the methodology and the understandability of the results, the authors are encouraged to perform a sensitivity analysis. Regression-based global sensitivity analysis, for example (Salas & Yepes, 2020), is an easy to implement, and not much time-consuming technique that provides valuable insights on the relative importance of the model input variables, which helps revealing whether the model is being sensitive to the more important of the research, such as the residents’ awareness of the need for mutual assistance.
3.2. Examination of the Method to explore a CDMP plan
In this sub-section, the authors present a methodology for analysing CDMP plans with the help of the evaluation tool presented in this paper. This is quite strange, since this sub-section is a part of the results section. This subsection should be therefore moved to the methodology section, and connect with the objectives/gaps of the introduction section. Perhaps the authors should think about rearranging their manuscript and change the focus of their paper from the development of a tool for assessing mutual assistance capacity, to the development of a methodology, consisting in a series of workshops, for evaluating CDMP plans and for raising resident’s awareness on the importance of mutual assistance for improving community response to huge earthquakes. Overall, the paper’s flow is bad, and I this way it can be greatly improved.
3.2.4. Questionnaire for data collection
On the other hand, the authors also performed a questionnaire to collect the necessary data for using the developed tool. This is very striking since the authors stated, in the methodology section, that “However, we judged that the actual data is unnecessary for a case study.”. The authors should revise the whole manuscript to fix all the inconsistencies. In the same vein, the authors include two results sections, in lines 330 and 446 of their manuscript, which reveals poor preparation and revision of their work.
In this section, the authors do not discuss, as expected, the results of the evaluation process (Table 2). Instead, they describe the process they have followed in conducting surveys both to verify the usability and issues of the developed tool (section 4.1.), and the usability and issues for creating a Community Disaster 62 Management Plans. I suggest the authors that they include the description of these surveys in the methodology, along with a justification of the need of these surveys (for testing the results of their method which consists in a series of workshops rather than in the quantitative evaluation of mutual-assistance capacity), and include the results of the surveys in the results sections. Then, in the discussion, they can employ the results of the survey to show how the process contributed to the achievement of the objectives (exploring the plans and raising awareness), but prior to this, it is necessary that the experiment is properly introduced to the reader.
- Conclusions
The authors claim that “This paper developed a tool to support exploring mutual assistance activities for promoting mutual assistance capability to earthquake disaster mitigation”, but it is very questionable that the developed tool is based on a valid methodology. As explained above, the authors do not provide in their paper information enough to evaluate the validity of their method for evaluating mutual assistance capacity. As they themselves acknowledge at the end of their conclusions, “From the point of validity view, the validity of the calculation method of the capability of mutual assistance and the simulation on mutual assistance activities should be improved.”. I therefore suggest that the authors focus on the methodology of workshops they propose for revising CDMP plans, and for raising residents’ awareness on the importance of mutual assistance.
The rest of statements in the conclusions, on the other hand, refers to the usefulness of the proposed tool for explaining the results to the residents, which has but little scientific soundness, and to the effectiveness of the set of workshops in raising residents’ awareness on the importance of mutual assistance. Here it lies, in my opinion, the main contribution of the reviewed work, and should be placed in a central point in the paper. It is therefore necessary that the authors identify the need of improving residents awareness of the importance of mutual assistance, as well as their understanding of CDMP plans, and that this could be done by means of an educational process, articulated via workshops in which the attendants have the support of digital tools such as that presented in this work to facilitate the illustration of the problem, and the understanding of the actions proposed in the CDMP plans.
- References in the review
- Salas, J., & Yepes, V. (2020). Enhancing Sustainability and Resilience through Multi-Level Infrastructure Planning. International Journal of Environmental Research and Public Health, 17(3), 962. http://doi.org/10.3390/ijerph17030962
Author Response
Dear reviewer 3
Thank you for your kind suggestion. I revised my manuscript as attached file.

Round 2
Reviewer 2 Report
The manuscript "A Methodology of Workshops to Explore Mutual Assistance 2 Activities for Earthquake Disaster Mitigation" by Kazuki Karashima, and Akira Ohgai, was significantly improved. It is now at the standard quality for publication.
The only suggestion for the Authors is to search for further improvements the English fluidity of the text where possible.
Author Response
Dear Reviewer 2
Thank you for your kind suggestion.
I attached a file to respond your suggestion.
Best regards,
Karashima

Reviewer 3 Report
Please find attached the comments to the revised manuscript. There is a comment from round one that was not addressed in the author's response, please provide an answer to it.
Kind regards

Author Response
Dear Reviewer 3
Thank you for your kind suggestion.
I attached a file to respond your suggestion.
Best regards,
Karashima
